# Coordinated Transcriptional Waves Define the Inflammatory Response of Primary Microglial Culture

**DOI:** 10.3390/ijms241310928

**Published:** 2023-06-30

**Authors:** Keren Zohar, Elyad Lezmi, Fanny Reichert, Tsiona Eliyahu, Shlomo Rotshenker, Marta Weinstock, Michal Linial

**Affiliations:** 1Department of Biological Chemistry, Institute of Life Sciences, The Hebrew University of Jerusalem, Jerusalem 91904, Israel; keren.zohar@mail.huji.ac.il (K.Z.); tsiona.e@mail.huji.ac.il (T.E.); 2Department of Genetics, Institute of Life Sciences, The Hebrew University of Jerusalem, Jerusalem 91904, Israel; elyad.lezmi@mail.huji.ac.il; 3Department of Medical Neurobiology, Institute for Medical Research Israel-Canada (IMRIC), Faculty of Medicine, The Hebrew University of Jerusalem, Jerusalem 91121, Israel; funarei@gmail.com (F.R.); shlomor@ekmd.huji.ac.il (S.R.); 4Institute of Drug Research, School of Pharmacy, The Hebrew University of Jerusalem, Jerusalem 91121, Israel; martar@ekmd.huji.ac.il

**Keywords:** innate immune system, interleukin, RNA-seq, purinergic receptor, ncRNA, inflammation, cytokines, TNF signaling

## Abstract

The primary role of microglia is to maintain homeostasis by effectively responding to various disturbances. Activation of transcriptional programs determines the microglia’s response to external stimuli. In this study, we stimulated murine neonatal microglial cells with benzoyl ATP (bzATP) and lipopolysaccharide (LPS), and monitored their ability to release pro-inflammatory cytokines. When cells are exposed to bzATP, a purinergic receptor agonist, a short-lived wave of transcriptional changes, occurs. However, only combining bzATP and LPS led to a sustainable and robust response. The transcriptional profile is dominated by induced cytokines (e.g., IL-1α and IL-1β), chemokines, and their membrane receptors. Several abundant long noncoding RNAs (lncRNAs) are induced by bzATP/LPS, including Ptgs2os2, Bc1, and Morrbid, that function in inflammation and cytokine production. Analyzing the observed changes through TNF (Tumor necrosis factor) and NF-κB (nuclear factor kappa light chain enhancer of activated B cells) pathways confirmed that neonatal glial cells exhibit a distinctive expression program in which inflammatory-related genes are upregulated by orders of magnitude. The observed capacity of the microglial culture to activate a robust inflammatory response is useful for studying neurons under stress, brain injury, and aging. We propose the use of a primary neonatal microglia culture as a responsive in vitro model for testing drugs that may interact with inflammatory signaling and the lncRNA regulatory network.

## 1. Introduction

Microglia act as the resident macrophages of the central nervous system (CNS). Their function is to maintain brain homeostasis and respond effectively to a broad spectrum of perturbations induced by acute stress from toxic agents or physical injury [1,2,3,4]. This is accomplished through communication with the surrounding neurons and astrocytes, and the release of signaling molecules such as ATP, that interact with specific purinergic receptors on the microglial membrane [5,6]. To protect neurons from apoptotic death and remove dying cells, microglia respond by changing their mode of activation and morphology [7,8,9]. In addition to oxidative stress and neuronal damage caused by aging, the release of pro-inflammatory cytokines is prolonged and exacerbates neurodegeneration [10]. Glial cells with excess amounts of neuroinflammatory markers (e.g., HLA-DR, CD68, and CD105) are abundant in neurodegenerative diseases such as Parkinson’s (PD) and Alzheimer’s (AD) diseases [11]. 

Cumulative evidence suggests that microglial inflammatory activity plays a role in age-dependent memory decline [12], synaptic plasticity [13,14], and neurodegeneration [15]. Cross-talk between microglia and neighboring cells (e.g., neurons, stem cells, blood vessels, and astrocytes) varies with age and brain regions [16]. Nevertheless, on a cellular level, it was shown that extracellular ATP triggers TNF-α release through purinergic P2 receptors, most likely the P2XR7 subtype, by activating multiple signaling pathways, including the ERK/p38 cascade [17,18]. Signaling through TNF-α and NF-κB pathways was studied in the context of inflammation and immunity [19]. While Tnfrsf1a, the main TNF-receptor, is widely expressed, Tnfrsf1b is primarily expressed in microglia. The affinity of TNF-binding determines downstream pathways such as JNK activation, NF-κB nuclear translocation, or PI3K/AKT recruitment [20,21,22,23]. 

The use of cellular models can benefit our understanding of the processes underlying brain injury and neuropathology [24]. Although in vitro cellular systems fail to reflect the complexity of the microenvironment of a living brain [25], primary microglia cultures from rodents [26] and humans [27,28] are attractive models for studying microglial diversity in neuropathological settings [24]. To better understand the processes involved in the regulation of microglial activity, studies have been performed under controlled conditions in isolated microglia prepared from neonatal rodents, cultured alone or with other cells (e.g., astrocytes) [29,30]. When exposed to activators such as lipopolysaccharide (LPS), interferon-gamma (IFNγ), and interleukins, these cells undergo transcriptional changes within a few hours, together with alterations in morphology [31], phagocytic ability, and inflammatory response [32,33]. 

In this study, we applied RNA sequencing analysis (RNA-seq) to quantify gene expression and examine the molecular changes occurring in a primary neonatal microglia culture. To better control the experimental setting and improve reproducibility, we did not add serum to the culture but provided proteins in the form of bovine serum albumin (BSA). We monitored the dynamics of the transcriptomic profile and cytokine release in response to LPS in the absence and presence of benzoyl ATP (bzATP), an agonist of the P2RX7 subtype of ATP receptors. The resulting gene expression signatures of mRNAs and ncRNAs highlight the regulatory network that governs the inflammatory and the full repertoire, and cellular responses.

## 2. Results

### 2.1. Functional Response of Primary Microglia

The function of the microglia was monitored by quantifying the secretion of TNF-α and IL-6 following activation with bzATP (Figure 1A). Prior to addition of bzATP secreted cytokines were below the level of detection (<5 pg/μg of cell lysate), but increased markedly at 8 and 24 h after stimulation. While the amount of IL-6 secreted was greater when collected over a period of 24 h than at 8 h (*p*-value < 0.0001), that of TNF-α was smaller (*p*-value < 0.001) (Figure 1A). Moreover, 8 h following stimulation, the absolute amounts of IL-6 and TNF-α secreted were 5.92 and 423.4 pg/μg of microglial cell lysate, respectively. Notably, these results were obtained from cultures supplemented with BSA instead of fetal calf serum (FCS). We showed that, following bzATP/LPS stimulation, the amounts of secreted proinflammatory cytokines significantly and reproducibly increased.

We then measured the transcript levels of Tnf and Il6 in untreated cells and upon bzATP/LPS stimulation. We noted that Tnf (encodes TNF-α) transcript levels can be detected in the naïve cells. This level increased significantly after 3 h and decreased after 8 h of bzATP/LPS. In quantitative terms, the Tnf transcript accounts for 0.02%, 1.2%, and 0.36% of all coding transcripts of untreated cells 3 and 8 h after bzATP/LPS, respectively. In contrast, the mRNA of Il6 (encodes IL-6) is below detection in naïve cells and its induction shows relatively slow kinetics (Figure 1B). IL-6 secretion displayed a continuous elevation, 24 h post-stimulus (Figure 1A) which agrees with the upregulation of the Il6 transcript, preceding protein secretion. We conclude that gene expression of TNF and IL-6 release is stimulus-dependent, and the kinetics of release confirms that each cytokine displays a distinct activation route [34].

### 2.2. A Short-Lived Transient Wave of Gene Expression by bzATP

To establish the microglial characteristics and quantify the molecular events that occur in culture, we first tested the cells’ transcriptome at 3 and 8 h after exposure to bzATP. Using RNA-seq analysis, we report that 10,835 genes were expressed that complied with the statistical quality threshold (see Section 4). Figure 2A shows the differential expression of each gene clustered by their fold change (FC) into nine expression patterns. For the definition of the expression trend (see Table 1). Figure 2B reports the number of genes in the presence of bzATP associated with each of the nine combined trends. The expression levels of almost all genes (96%) are unchanged after 3 h (labeled ‘Same’ Figure 2A). Notably, most genes were still marked as unchanged after 8 h (labeled ‘Same–Same’; Figure 2B). Applying strict thresholds of false discovery rate FDR (q-value) < 0.05 and a minimal expression level of >10 TMM confirmed that the vast majority of the filtered DE genes (93%) are labeled ‘Same’, with only 16 genes (0.5%) significantly changing their expression in a monotonic trend (i.e., labeled Up–Up and Down–Down; Appendix A). 

The downregulated (seven genes, FDR q-value ranges from 3.1 × 10^−18^ to 2.9 × 10^−10^; Figure 2C) and upregulated DE genes (nine genes, FDR q-value ranges from 2.7 × 10^−18^ to 5.6 × 10^−4^; Figure 2D) are shown. Those continuously downregulated contribute to cell migration and differentiation (e.g., Hyal1 and Sema4d). In contrast, inflammatory signals dominate the overexpressed genes in Figure 2D. For example, Trem1 is a scaffold membrane protein that acts in chemotaxis and regulates the killing of Gram-negative bacteria. Additional genes belong to chemokines (Cxcl2, chemokine C-X-C motif ligand 2) and their immediate pathways. Thus, the induced genes revealed a distinctive set of immune-related response sensors in microglia. We conclude that exposing murine microglia to bzATP led to a transient wave of gene expression that faded out over a longer timeframe (8 h). 

In addition to DE genes with a monotonic trend, we investigated genes with a slow kinetics (i.e., only significantly changed at 8 h). Appendix A lists 26 of the most significant such genes (at a strict FDR < 1.0 × 10^−10^; TMM > 10). Downregulated genes include those involved in trafficking (e.g., Rab7b, Snx7) and cell–cell interaction (e.g., Cadm1, Vwf). The larger fraction of DE genes with slow kinetics is associated with the regulation of immune maintenance via TNF signaling. Tnfrsf1b (tumor necrosis factor receptor superfamily, 1b) and Ltb (lymphotoxin-beta) promote cytokine cell surface signaling, whereas Itgam directly activates TNF-primed neutrophils, regulates neutrophil migration, and is involved in the production of superoxide ions in microglia (Appendix A). We conclude that the microglial culture increased its immune responsiveness, but the fold change in expression for the slow kinetic genes is minimal (<2-fold; Appendix A).

### 2.3. Global Alteration of the Cell Transcriptome by bzATP/LPS Is Synchronized and Long-Lasting

We examined the coherence of the transcriptional programs in each experimental group, by performing PCA analysis (Figure 3A). The results show that the five groups are tightly clustered, with negligible variance within each group relative to the variance among the tested experimental conditions (i.e., N.T., bzATP, and bzATP/LPS at two time points). The PCA results emphasize the strong and persistent effect induced by bzATP/LPS. The variance explained by the two principal components (PC1, PC2) is 68.6%. 

Figure 3B shows the partitioning of the DE genes into nine expression patterns in the presence of bzATP/LPS. In contrast to the results for bzATP alone (Figure 2B), bzATP/LPS drastically changed the expression of a large fraction of the genes (44.3%) within 3 h (1924 and 2851 for Up- and Down-labeled genes, respectively; Figure 3B). The expression profiles of all 10,769 expressed genes before (time = 0), 3, and 8 h after treatment are shown in Appendix A. The experimental results are listed in Appendix A.

### 2.4. Expression of Interleukins as Indicators of Microglial Response

The function of the microglia was tested by measuring the release of pro-inflammatory cytokines (Figure 1A), with interleukins (ILs) and their receptors serving as indicators of the inflammatory state. We conducted a semi-quantitative RT-PCR assay to test the kinetics and extent of gene expression for major interleukins and their receptors. Figure 3C further substantiates the slow kinetics observed for Il6. Furthermore, a strong increase in the expression levels of Il1a and Il1b in the presence of bzATP/LPS is evident. In contrast, the levels of expression of Tnf and Tnfrsf1b are already substantial prior to cell activation.

We expanded the analysis to display the relative expression of all IL-gene transcripts (33 expressed genes, collectively called the IL-gene set; Appendix A). Figure 3D shows that both Il1a and Il1b transcripts increase in the presence of bzATP. While Il1b transcripts account for only 2% of the IL gene set (assuming TMM of the IL-gene set is 100%) in the presence of bzATP and LPS for 3 h, transcript levels increase > 2000-fold (Figure 3D), accounting for 35% of the mRNAs of the IL-gene set. The kinetics of Il1a and Il1b are very similar (Figure 3C). In particular, bzATP/LPS increased the level of Il1a mRNA dramatically (~500-fold in 3 h), accounting for half of the total number of transcripts in the IL-gene set. Unlike interleukins, the receptor Il1rl1 increased nine-fold 3 h after activation by BzATP/LPS with no further increase at a later time point (8 h; Figure 3D). The short-lived wave of bzATP activation is reflected by the expression of IL receptors. The Il1rn (interleukin 1 receptor antagonist), which inhibits the activities of IL-1α and IL-1β, was induced transiently by bzATP (1.8-fold, 3 h) but returned to its baseline levels at 8 h. The addition of LPS led to a substantial and permanent increase in its expression (13.1 and 10.7-fold, at 3 and 8 h, respectively, Figure 3D). The expression levels of Ils and their receptors shed light on the specific roles of some but not other Ils. For example, Il17ra (interleukin 17 receptor A) accounts for 30% of transcripts of the IL gene set in untreated cells. However, exposure of the culture to bzATP or bzATP/LPS only slightly affected its expression (~25%; Figure 3D), in agreement with the potential role of Il17 signaling in disease progression rather than in acute inflammatory signaling.

### 2.5. Changes in Gene Expression by Orders of Magnitude Drive the Cellular State

To assess the kinetics and extent of the transcriptional waves, we focused on the DE genes according to the fold ratio in the presence of bzATP/LPS relative to untreated cells. Figure 4 shows a sorted list of 25 such genes. Already after 3 h, the degree of upregulation is substantial and remains high for most genes at 8 h (Figure 4A,B). The upregulation ranges from 5.5- to 2800-fold (Figure 4C). The majority of the transcripts encoded by these genes are functionally and physically connected (Appendix A). Among the highly connected proteins are Tnf and Il1b, and components of chemokine signaling (Cxcl10, Ccl2, Cxcl2, Ccl5, Csf2). Among the downregulated genes (>10 TMM; 720 genes; a factor of 2 to 20), no significant gene ontology (GO) or KEGG pathways is evident. However, of a special interest is the P2RX7, an ATP-gated cation channel that participates in neuroinflammation and pathophysiological processes. The expression of the P2rx7 transcript was downregulated by 8.3-fold at 3 h after the addition of bzATP/LTP and partially recovered at 8 h. 

### 2.6. Enrichment Analysis of the Transcriptome Induced by bzATP/LPS 

Figure 5A shows the enrichment for the genes upregulated by bzATP/LPS according to KEGG pathways. The partition of these genes (total 1924) by the expression trend of Up–Up is shown. The monotonically increasing genes (Up–Up; 85) are associated with cytokine pathways (e.g., TNF signaling), viral infection, and inflammation-based diseases. The enrichment analyses for all nine expression trends are available in Appendix A. The KEGG pathway view confirmed the occurrence of a coordinated transcriptional wave by bzATP/LPS. We further analyzed the upregulated DE genes according to the gene ontology (GO) annotations (Figure 5B). Inspection of the molecular functions revealed a strong enrichment for cytokine activity and cytokine receptor binding, inflammation and extracellular localization. Repeating the GO enrichment test using a more relaxed threshold (*p*-value < 1.0 × 10^−7^) further supported the annotations of cytokine regulation and kinase phosphorylation (Appendix A). We conclude that a global gene expression view supports a robust and long-lasting induction of cytokines and chemokines.

### 2.7. Inflammatory-Related Signaling Pathways Are Significantly Induced by Exposing Cells to LPS 

We compared the pathway characteristics in cells exposed to bzATP relative to cells exposed to bzATP/LPS by creating a STRING protein–protein interaction (PPI) map [35]. For each PPI map, we projected the most significant 100 DE genes (sorted by *p*-value). Figure 6 displays the difference in the resulting network for cells exposed to bzATP/LPS conditions. Interesting, the connectivity for cells exposed to bzATP is low and the association with any of the pathways is negligible. In contrast, a strongly connected network was observed following the bzATP/LPS condition (Figure 6A) with multiple inflammatory signals including TNF and NF-κB signaling (FDR = 5.48 × 10^−9^), IL-1 and IL-3 signaling (FDR = 1.01 × 10^−5^), and oxidative damage response (1.49 × 10^−5^) listed for the bzATP/LPS condition (Figure 6B). The WikiPathways [36] results for the 100 most significant genes are shown for bzATP and bzATP/LPS (Appendix A). We concluded that bzATP/LPS (8 h) changes gene expression followed by numerous inflammatory pathways, most notably TNF-α and NF-κB signaling.

### 2.8. Alternative Signaling Patterns Dominate the Microglia Activation States

The strength of an in vitro cellular system is determined by the specificity of the response. We reanalyzed the DE genes while focusing on the PI3K-Akt branch from KEGG [37]. Figure 6B highlights the differences in the expression. The activation of major signaling PI3K-Akt is evident (Figure 6B, red), with the involvement of p38 and NF-κB, but not IFNβ. The most activated KEGG pathways are the TNF (Appendix A) and and NF-κB signaling pathways (Appendix A).

### 2.9. Stimulation of Microglia by bzATP/LPS Affects the Expression of Abundant ncRNAs 

In addition to the identified coding genes, there are 1112 non-coding (ncRNA)-expressing genes (Figure 7A). The partitioning of these sets revealed that lncRNA, antisense, processed pseudogenes, and TEC (to be experimentally confirmed) occupied the majority of the RNA biotypes (Figure 7B, left). The most expressed group that accounts for 65% of all expressed ncRNAs includes a few miRNAs, small nucleolar and nuclear RNAs (snoRNAs and snRNAs, respectively), followed by lncRNAs (13%; Figure 7B, right). Among the most abundant ncRNAs are Malat1, mitochondrial rRNAs (Rnr1 and Rnr2), Neat1, Bc1, and others.

Figure 7C shows the subset of genes that are abundant (>25 TMM) and whose DE have at least a two-fold difference in expression following 8 h treatment with bzATP/LPS relative to untreated cells. Appendix A lists all significant DE ncRNA genes. The function of most genes is unknown but most were upregulated (60%). Interestingly, those related to inflammation dominate among the DE ncRNA genes. For example, Il1bos (interleukin 1 beta, opposite strand) was induced >80-fold, in agreement with the strong change in expression of the sense Il1b gene (Figure 4C). 

## 3. Discussion 

Microglia are characterized by their ability to sense and react to abnormal and disturbing conditions [38]. Because of their crucial importance in maintaining brain homeostasis, the amplitude of the change in their gene expression must be tightly controlled. While hundreds of genes are up- and downregulated in response to the stimuli, those that are upregulated by orders of magnitude drive the microglia culture’s robust inflammatory response. In vivo, the activation cascade of microglia begins with the binding of extracellular molecules to Toll-like receptors [39], thereby resulting in changes in cytokine secretion, cell proliferation, migration, and phagocytosis [40]. Microglia respond by sensing the local concentrations of molecules, including amyloid β (Aβ), nucleic acids, selective estrogen receptor modulators [41], reactive oxygen species (ROS), ATP, and more. In the injured brain, a switch in microglial activity [42,43] occurs following the release of ATP from dying astrocytes and neurons [44]. Microglia also release ATP through exocytosis [5], thus creating a feedback loop that modulates the cells’ activation program. Exposure of microglia to bzATP induced a transient wave of DE genes, including Ccl2 (chemokine C-C motif ligand 2, Figure 4), that is upregulated in the brains of AD patients [45]. In mouse models of AD, overexpression of Ccl2 was shown to cause microglia-induced amyloid β oligomerization, worsening of tau pathology, and consequently led to an increase in IL-6 release [46]. This study examined the molecular characterization of neonatal primary microglia culture to external stimulants in a defined environment, without the addition of undefined factors from serum. It was shown that, while microglial maturation occurs weeks after birth, the addition of factors from serum to the culture perturbs the cells’ morphology and function (e.g., phagocytic capacity), and makes the culture prone to non-reproduceable results [30]. 

Here, we highlight the contribution of abundant lncRNAs in establishing the inflammatory response. For example, a knockout mouse model of the Morrbid gene (myeloid RNA regulator of BCL2L11 induced cell death) resulted in decreased monocytes, increased apoptosis, and increased sensitivity to infection. The Ptgs2os2 (prostaglandin–endoperoxide synthase 2, opposite strand 2, also called Gm26687 or Linc-Cox2) is dynamically regulated and controls the immune gene expression in response to LPS. Based on Ptgs2os2-deficient mice, it was shown that Ptgs2os2 acts as an enhancer for the neighboring gene prostaglandin–endoperoxide synthase (Ptgs2). In our system this gene was strongly upregulated already 3 h after bzATP/LPS (Figure 4C). Ptgs2os2 lncRNA also act to regulate the innate immune genes in vivo [47]. We suggest that several of the master regulators of the inflammatory response in the primary neonatal microglia culture are lncRNAs. 

We observed an exceptional coherence in the levels of gene expression within each experimental condition group (Figure 3A). Accordingly, the duration and nature of the stimulus (e.g., bzATP, LPS, IFNγ, TNF, growth factors, and their combinations) govern the outcome (e.g., TNF and NF-κB, Appendix A, respectively). Similar results were generated from primary microglia cultured from rats, confirming distinct molecular signatures that specify LPS stimulation, pro-inflammatory cytokines (e.g., IFNγ and TNF-α) and the sequential introduction of anti-inflammatory cytokines [26]. The level of Tnf transcript increased considerably by LPS (65-fold). However, the upregulation in TNF transcripts preceded the release of the TNF proteins (Figure 1). A similar trend was detected for IL-6. We propose that transcriptional wave kinetics is an important determinant of the cellular response. For example, Tnfrsf1b (but not Tnfrsf1a) was strongly upregulated at 3 h by bzATP/LPS (4.9-fold; Appendix A). The differential expression of the different TNF receptors dictates TNF binding specificity. While the soluble TNF form is limited to the Tnfrsf1a gene product, the membranous form has no preference towards Tnfrsf1a and Tnfrsf1b (Appendix A). A similar trend was applied for the NF-κB signaling. Tnfaip3 was upregulated by the addition of bzATP/LPS (3 h, 14.6-fold, Appendix A) but returned to a baseline level after 8 h. Tnfaip3 is a gatekeeper of inflammation through the suppression of NF-κB activation [48]. By introducing bzATP/LPS in the absence of an undefined component from serum, the signaling cascade of NF-κB is manifested via the MAP kinase signaling (i.e., the upregulation of p38 that is encoded by Mapk11-Mapk14). The other components are mostly unchanged (marked gray, Figure 6B). We conclude that the nature of the stimulus in cultured neonatal microglia differs not only in its kinetics but also in the signal transduction cascade leading to cell response. While only 16.5% of the genes overlap between the TNF and NF-κB signaling pathways (based on the KEGG gene list), it seems that the nature of the stimuli governs the extent of the inflammatory signaling cascade (Appendix A).

Previous studies have used microglial cell lines (e.g., N9, BV2) to investigate microglial biology in vitro. However, comparing the gene expression signature from adult microglia to primary culture, cell lines, and other monocyte preparations showed the similarity of the cultured primary microglia to adult microglia, whereas this signature was not detected in any of the microglial cell lines tested [49]. Indeed, the transcriptomes of BV2 and the primary microglial cells are substantially different [50,51], and do not accurately represent primary microglia [52]. In the effort to identify genes and pathways that are shared across different neuropathologies, a list of core genes (total 86) was compiled that discriminates reactive from homeostatic conditions [53]. Recently, single cell analysis was applied on post mortem human brains [25] to yield a rich catalog of markers for cell identity (e.g., homeostatic, proliferating, immune responsive). These marker genes were quantified with respect to microglial age and neurodegenerative diseases [54]. We provided evidence for a coordination in gene expression that mimics the immune-response axis. A comparison of microglial responses (8 h in bzATP/LPS) with LPS and IFNγ [55] (Appendix A) did not substantially changes the overall response. Moreover, in an in vivo model in which LPS was injected into mice and the expression changes in isolated microglia were monitored [56], the strongest effect was associated with the TNF inflammatory response. 

Despite the lack of communication with other cell types (e.g., neurons, astrocytes, stem cells), the microglial cultures exerted a rich repertoire of responses to external stimuli [57]. Such a system is attractive for revealing the underlying mechanism of anti-inflammatory drugs in the ongoing effort to control pathophysiological conditions of the CNS at the cellular level [58]. For example, ladostigil prevented the development of age-related memory deficits in aged rats by a combination of immunomodulatory and antioxidant effects [59]. It was further shown that ladostigil mediated its protective effects through suppression of microglia [60]. Different noxious stimuli such as injury, hypoxia, and stress conditions are associated with rebalancing inflammatory cytokines (e.g., TNF-α, IL-6, and IL-1β). Based on our results, we propose the cellular system as an attractive setting to test reagents and drugs that interact with components of the innate immune system and inflammatory signaling. 

## 4. Materials and Methods

### 4.1. Compounds and Reagents

Dulbecco’s Modified Eagle Medium (DMEM), DMEM/F12, gentamycin sulfate and L-glutamine were obtained from Biological Industries (Beit-Haemek, Israel), and 2′-3′-*O*-(4-benzoyl benzoyl) adenosine 5′-triphosphate (bzATP), bovine serum albumin (BSA), and LPS were purchased from Sigma-Aldrich (Jerusalem, Israel). 

### 4.2. Preparation of Microglial Cultures

Primary microglia were isolated from the brains of neonatal male Balb/C mice (Harlan Sprague Dawley Inc., Jerusalem, Israel) as described in [61]. The cells were isolated and plated in poly-L-lysine-coated flasks for one week. Following an enzymatic dissociation protocol, the non-adherent and loosely adhered cells were re-plated for 1 h on bacteriological plates, which allowed sorting out cells exhibiting a slower kinetic of adherence. Microglial cells were propagated by supplementing the culture with 10–20% of medium conditioned from L-cells that produce mouse-CSF (colony-stimulating factor). Under such conditions, the microglial culture remains responsive for four weeks. Before conducting experiments, the microglial cells were removed from the conditioned medium for 24 h. In all experiments the heat-inactivated fetal calf serum (FCS) was replaced by purified BSA. The purity of microglia was confirmed by immunostaining and morphological criteria to distinguish them from astrocytes and oligodendrocytes. Distinct morphology and staining by P2Y12, F4/80, CR3 (complement receptor-3), and galectin-3/MAC-2 validated the purity of the culture to be >95% as shown in [62].

### 4.3. Measurement of Cytokines

Activation of the cultured microglia was measured by cytokine release as previously described [63] and according to manufacturer’s protocols. Cells were grown to 75% confluence in 6-well plates. Measurements of cytokine secretion were made 8 and 24 h after activation in the presence of BSA (0.4 µM) using Max deluxe (Biolegend, San Diego, CA, USA) ELISA kits [63]. We calibrated the use of LPS. Data presented are for microglial cells stimulated by bzATP (400 µM) only, LPS (1 µg/mL), and their combination (bzATP/LPS). The protein content of the cells was measured using BCA Protein Assay (Pierce, Meridian, Rockford, IL, USA)). Assays were repeated with an internal control for cytokines for 5 × 10^5^ cells per well. Cells were harvested by scraping with a rubber policeman, washed with PBS (4 °C), and counted. We obtained 40 μg lysate from 5 × 10^5^ cells following lysis and clearance by centrifugation (14,000× *g*, 10 min, 4 °C). Each cytokine assay was calibrated by an internal standard curve.

### 4.4. RNA-Seq

Microglial cultures were harvested using a cell-scraper. Total RNA was purified from ~10^6^ cells using QIAzol Lysis Reagent RNeasy plus Universal Mini Kit (QIAGEN, GmbH, Hilden, Germany). To ensure homogenization, a QIAshredder (QIAGEN, GmbH, Hilden, Germany) mini-spin column was used. Samples were transferred to a RNeasy Mini spin column and centrifuged for 15 s at 8000× *g* at room temperature. The mixture was processed according to the manufacturer’s standard protocol. Samples with an RNA integrity number (RIN) > 8.5, as measured by Agilent 2100 Bioanalyzer, were considered for further analysis. Total RNA samples (1 μg RNA) were enriched for mRNAs by pull-down of poly(A)^+^ RNA. RNA-seq libraries for RNA > 200 nt were prepared as described [64]. We used KAPA stranded RNA-seq kit (Roche, Basel, Switzerland) according to the manufacturer’s protocol and sequenced using Illumina NextSeq 500 to generate 85 bp single-end reads. 

### 4.5. Gel Based RT-PCR

Reverse transcription polymerase chain reaction (RT-PCR) assays followed by a gel-based separation was performed for selected genes. cDNA was prepared by the High Capacity cDNA Reverse Transcription Kit (ThermoFisher, Waltham, MA, USA) according to the manufacturer’s instructions, using RT random primers and MultiScribeTM Reverse Transcriptase. For PCR reaction we used the PCRBIO-HS Taq mix (PCRBiosystem; London, UK) according to the recommended protocol that consists of a denaturation step at 95 °C (2 min) followed by 35 cycles (10 s at 95 °C, 15 s annealing at 60 °C, and 10 s extension for at 72 °C). The tested genes are: Il6 (NM_031168), Fw: 5′CACTTCACAAGTCGGAGGCT; Rev: 5′GGAGAGCATTGGAAATTGGGG (380 nt); Tnf (NM_013693), Fw: 5′ACAGAAAGTCATGATCCGCGA; Rev: 5′GTTTGCTACGACGTGGGCT (288 nt); Il1a (NM_010554), Fw: 5′AGGGAGTCAACTCATTGGCG; Rev: 5′ACTTCTGCCTGACGAGCTTC (449 nt); Il1b (NM_008361), Fw: 5′TGCCACCTTTTGACAGTGATG; Rev: 5′GGAGCCTGTAGTGCAGTTG (351 nt); Tnfrsf1b (NM_011610), Fw: 5′CACTTGGGGCCGACTTGTTA; Rev: 5′CCGTCTCCTTCCCACAACAC (445 nt). β-actin, Fw: 5′CTGGAACGGTGAAGGTGACA; Rev: 5′AAGGGACTTCCTGTAACAATGCA (173 nt). The intensity of β-actin was used as a control. 

### 4.6. Bioinformatic Analysis and Statistics

All next-generation sequencing data underwent quality control using FastQC, version 0.11.9 [65] and were processed using Trimmomatic, version 0.39 [66], aligned to GRCh38 using STAR, version 2.7.6a. All genomic loci were annotated using GENCODE version 32 [67]. Trimmed mean of M-values (TMM) normalization of RNA read counts and differential expression analysis were performed using edgeR, version 3.36.0 [68]. TMM is a between-sample method that is suitable to compare different libraries. The low variability of the TMM values within a triplicate group confirms the quality of the RNA-seq data. For differential expression (DE) analysis, additional filtrations with FDR q-value < 0.05 and a minimal expression level of >10 TMM for the average of naïve and bzATP treated cells were applied. The term ‘Same’ marks changes in expression that are bounded by 50% (i.e., at a 0.67 to 1.5-fold difference relative to genes expressed in naïve cells). The partition of DE genes to clusters was done according to threshold listed in Table 1.

Pathway and gene-set enrichment analyses were performed using the clusterProfiler: enrichKEGG, version 4.2.0 program and pathview for visualization. GO annotation enrichment was based on GOrilla statistical tool [69]. Protein–protein interaction map was based on high scoring STRING interactome (score > 0.9) [35]. ID conversion from Ensemble to Entrez was carried out by the annotation package (biomaRt, org.Mm.eg.db, version 3.14.0). Experiments contained a minimum of three biological replicates. The cytokine quantification data were analyzed by one-way analysis of variance (ANOVA), using IBM SPSS Statistics Version 19 followed by Duncan’s post hoc test. Results from microglial cell experiments are presented as mean ± SD (standard deviation). All other statistical tests were performed using R-base functions. When appropriate, *p*-values < 0.05 were calculated and considered statistically significant. Principal component analysis (PCA) was performed using the R-base function “prcomp”. The analysis captures the maximum amount of variation in the data, visualized in two dimensions. PC1 explained the most variation and the subsequent component (PC2) explained the second most informative data variation. Figures were generated using the ggplot2 R package, version 3.3.5.

## Figures and Tables

**Figure 1 ijms-24-10928-f001:**
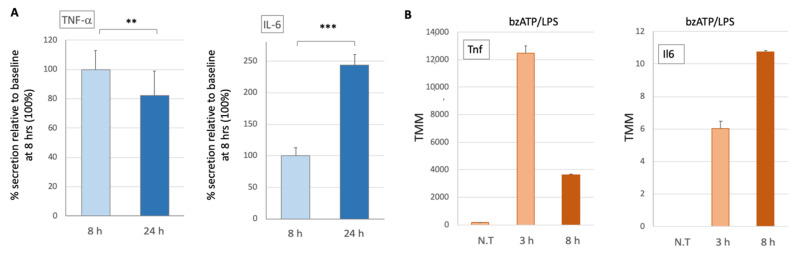
Quantitation of TNF-α and IL-6 released from primary neonatal murine microglial culture following stimulation. (**A**) Kinetics of TNF-α and IL-6 proteins upon bzATP/LPS stimulation. Values are normalized to protein concentrations (μg lysate) from the adhered cultured cells. The levels of TNF-α and IL-6 were below detection in unstimulated cells. Samples for the study were taken from conditioned media supplemented with BSA, harvested, and measured 8 and 24 h after stimulation. A mean and standard deviation (s.d.) of 4 experiments are shown (each group, n = 22–24). Statistical significance marked by asterisks implies the results of the Mann Whitney test *p*-values of 0.001 (**) and <0.0001 (***). (**B**) Kinetics of Tnf and Il6 transcripts upon bzATP/LPS stimulation based on RNA-seq data. Results are the average of biological triplicates. N.T., not treated.

**Figure 2 ijms-24-10928-f002:**
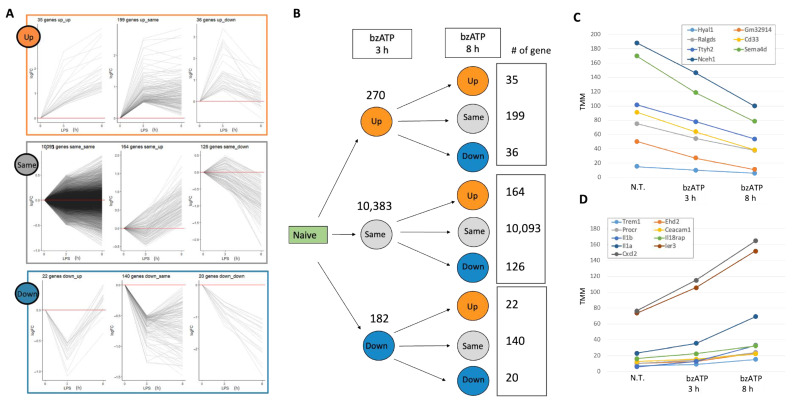
Trends in gene expression of microglial genes in the naïve cells (not treated, N.T.) and following activation protocol with BzATP for 3 h and 8 h. (**A**) The 10,835 identified differentially expressed genes are classified into nine groups based on their combined expression trend (Up, Down or Same, and their combinations; see Table 1). The baseline (logFC = 0) is shown by horizontal red line. (**B**) Summary statistics of the differentially expressed (DE) genes following activation with bzATP. The partition of all expressed genes (10,835) by their expression trend. Cells were exposed to bzATP and tested for their gene expressed at 3 h relative to naïve cells and at 8 h relative to 3 h. The number of genes labeled as Up, Same, Down is indicated. (**C**) The DE genes with a continuous downregulation (labeled Down–Down) from N.T. to 3 h and 8 h of bzATP treatment. (**D**) A set of upregulated genes (labeled Up–Up). The analysis is based on the data in Appendix A.

**Figure 3 ijms-24-10928-f003:**
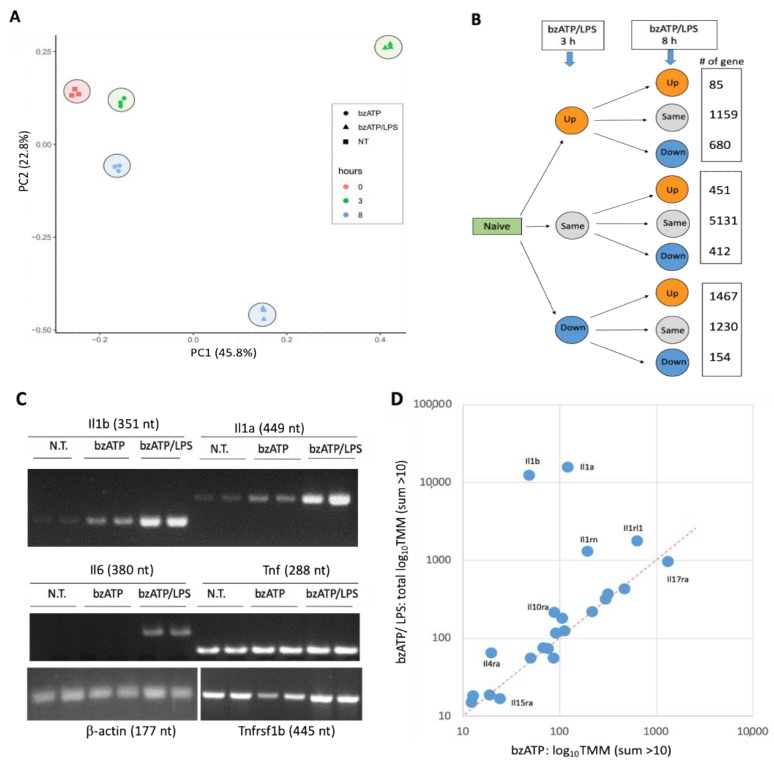
Global expression trends by RNA-seq and interleukin gene set expression profile. (**A**) PCA analysis of the 15 RNA-seq samples. (**B**) Gene expression levels of untreated microglial culture and following bzATP/LPS for 3 h and 8 h. A partition of all 9 expression trends along with the number of genes associated with each trend is shown (total for 10,769 genes). For details, see Appendix A. (**C**) RT-PCR results on untreated cells and cells after 8 h exposure to bzATP and bzATP/LPS (in biological duplicates). β-actin serves as an internal control. (**D**) Analysis of 33 IL-gene set activated by bzATP (*x*-axis) and bzATP/LPS after 8 h (*y*-axis). Only genes with mean expression levels of >10 TMM are shown.

**Figure 4 ijms-24-10928-f004:**
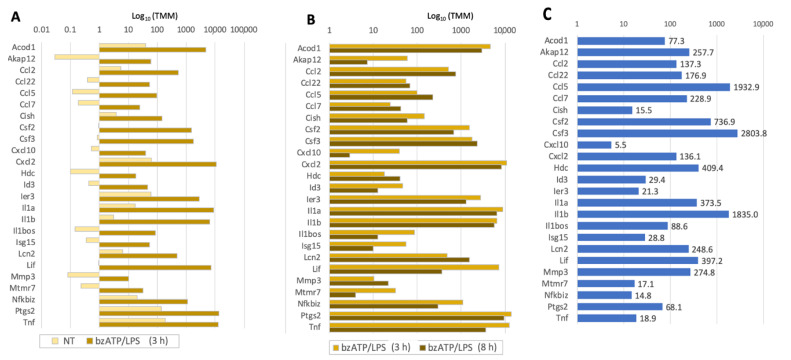
The top 25 induced genes sorted by their extent of induction by bzATP/LPS. (**A**) log_10_TMM of the expression in naïve, non-treated cells (NT) and 3 h of bzATP/LPS. (**B**) log_10_TMM of expression after 3 and 8 h of bzATP/LPS stimulation. The genes in (**A**–**C**) are listed alphabetically. (**C**) Increases after 8 h relative to that in untreated cells, e.g., Csf3 (colony stimulating factor 3) increases 2804-fold. See Appendix A.

**Figure 5 ijms-24-10928-f005:**
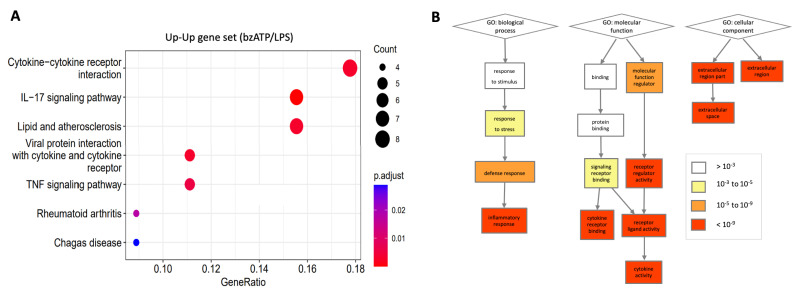
Functional annotation enrichment. (**A**) KEGG pathway enrichment for DE genes that were monotonically upregulated (i.e., Up–Up). Statistical enrichment of *p*-adjust is depicted by the colors (blue to red) for FDR < 0.05. The size of the dots depicts the number of proteins. Gene ratio depicts the fraction of genes in the cluster that are included in a specific pathway, *x*-axis. (**B**) Enrichment analysis for GO annotations of upregulated DE genes. Analysis performed on strongly upregulated genes (total of 585 genes, >1.5-fold increase in expression relative to non-treated cells, average expression level > 10 TMM). The enrichment results for GO biological process, molecular function, and cellular component satisfied by *p*-value < 1.0 × 10^−9^ (red color). Color codes of white to red indicate the adjusted *p*-values after multiple testing corrections.

**Figure 6 ijms-24-10928-f006:**
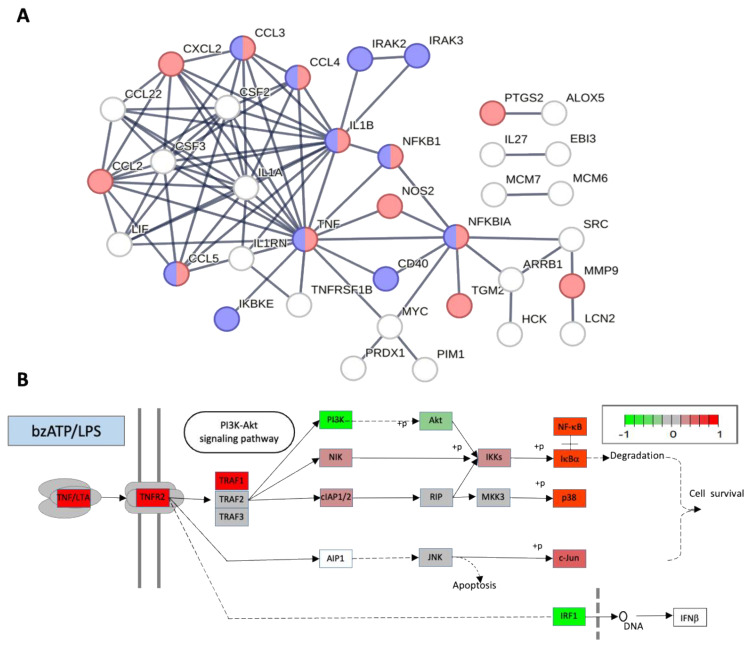
Functionally enriched pathways from RNA-seq results. (**A**) STRING interaction network for the 100 most significant DE coding genes at 8 h of RNA-seq results of bzATP relative to bzATP/LPS, sorted by the false discovery rate (FDR) *p*-values. Results are from STRING high confidence connections (>0.95) with at least two connected genes. PPI enrichment *p*-value < 1.0 × 10^−16^. In red and blue are genes annotated by WikiPathway [36] as IL-18 signaling (FDR enrichment 1.0 × 10^−12^), and Toll-like receptor signaling (FDR enrichment 1.5 × 10^−9^), respectively. Appendix A lists the top 100 analyzed genes. Appendix A shows the statistical results of significant enriched pathways from WikiPathway that meet FDR with a statistically significant FDR of <1 × 10^−2^. (**B**) PI3K-Akt pathway from the KEGG pathway map of TNF signaling labeled by the DE gene pattern. Microglia cultures treated with bzATP/LPS for 8 h. Genes are colored in four categories according to their expression trends. The colors indicate genes that are downregulated (green), upregulated (red), unchanged (gray), or not detected (white) with respect to their expression levels in untreated culture.

**Figure 7 ijms-24-10928-f007:**
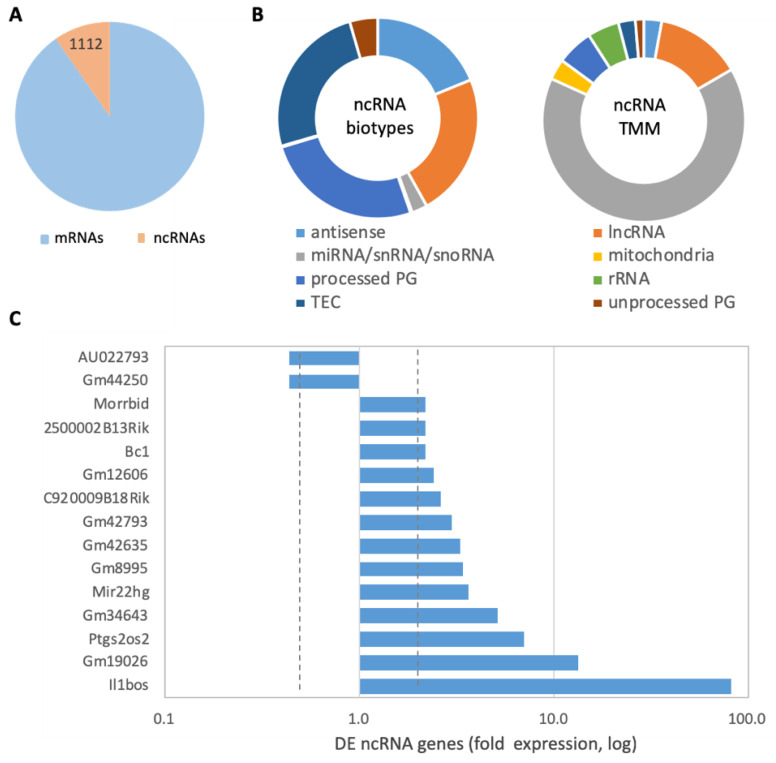
Expression of ncRNAs in cells stimulated by bzATP/LPS. (**A**) Partition of coding and ncRNA expressing genes. (**B**) The types of molecules (left) and their amount (TMM, right). (**C**) DE ncRNA genes with 2 downregulated and 13 upregulated genes, following 8 h with bzATP/LPS relative to untreated cells. Vertical dashed lines show 2-fold change in expression. All genes are expressed with >25 TMM and the DE FDR q-value range from 2.0 × 10^−17^ to 2.0 × 10^−30^. For DE analysis of ncRNA, see Appendix A.

**Table 1 ijms-24-10928-t001:** Thresholds used for assigning DE genes for 9 clusters with distinct expression trends.

FC1 3 h vs. N.T.	Trend FC1 ^a^	FC2 8 h vs. 3 h	Trend FC2	Combined Trend
log(FC1) > 0.5	Up	log(FC2) ≥ log(FC1) + 0.5	Up	Up–Up
log(FC1) > 0.5	Up	log(FC2) ≤ log(FC1) + 0.5 and log(FC2) ≥ log(FC1) − 0.5	Same	Up–Same
log(FC1) > 0.5	Up	log(FC2) < log(FC1) − 0.5	Down	Up–Down
−0.5 ≤ log(FC1) ≤ 0.5	Same	log(FC2) > log(FC1) + 0.5	Up	Same–Up
−0.5 ≤ log(FC1) ≤ 0.5	Same	log(FC2) ≤ log(FC1) + 0.5 and log(FC2) ≥ log(FC1) − 0.5	Same	Same–Same
−0.5 ≤ log(FC1) ≤ 0.5	Same	log(FC2) < log(FC1) − 0.5	Down	Same–Down
log(FC1) < −0.5	Down	log(FC2) ≥ log(FC1) + 0.5	Up	Down–Up
log(FC1) < −0.5	Down	log(FC2) ≤ log(FC1) + 0.5 and log(FC2) ≥ log(FC1) − 0.5	Same	Down–Same
log(FC1) < −0.5	Down	log(FC2) < log(FC1) − 0.5	Down	Down–Down

^a^ FC1 refers to differential expressed genes relative to naïve cells at 3 h; FC2 refers to differential expressed genes at 8 h relative to the results at 3 h. N.T., not treated.

## Data Availability

RNA-seq data files were deposited in ArrayExpress under the accession E-MTAB-10450. All other data is available through the Appendix A.

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
