# Peer review of "Coordinated Transcriptional Waves Define the Inflammatory Response of Primary Microglial Culture"

_ijms, 2023, doi:10.3390/ijms241310928_

Round 1

Reviewer 1 Report

Review on the manuscript titled “Coordinated transcriptional waves and changes in abundant
lncRNAs define the inflammatory response of primary microglial culture” by Zohar et al., 2023.

                Microglia is the primary agent inducing immune response based on the afferents evoked my various substances and markers. It’s been repeatedly shown that microglial signaling is a primary source of consequent response cascades in other brain cell types. Since immune response is evoked with a large range of events, including shifted homeostasis of the brain regions as in the case of AD, or oxidative stress in case of ischemia, the adequate model of microglia-mediated immune response in brain is highly appreciated.

                The main advantage of the manuscript besides composite agent proposed as an inflammatory agent, the authors also performed the temporal assessment of the immune response. 

                They also addressed ncRNA impact in the course of inflammatory cascades, and finally, displayed temporal course of microglial DEGs response dynamics patterned in oscillated manner, as depicted in Supplementary Material.

As a conclusion, the authors “propose the use of a primary neonatal microglia culture as a responsive in vitro model for testing drugs that may interact with inflammatory signaling and the ncRNA regulatory network”.

The study is well designed and the conclusion bears high value in pursuing inflammation-tackling methods and targets. The manuscript is transparent and is easy to read. The results should be actual for the researchers in the field.

Several notes on the manuscript,

1)      it would be worth providing some putative (drug) targets (genes) at the conclusion section, if that’s not a closed info. The analysis of temporal dynamics of alternative splicing events would be also quite relevant within the scope of the paper.

2)      FDR (q-value) < 0.05 and >10 TMM  - TPM  (s. 213)

3)      I do not quite comprehend the ‘same’ category based on the Fig. S1. How could ‘same’ be in the first position? We’re talking of DEGs, so the genes cannot keep the initial expression rate. If it means that on the first 3 hours there’s not statistically significant expression change (?), then what does the ‘same-same’ term imply? The picture legend probably can be extended with some elaboration.

4)      Though the Table S1 acronyms are decoded in the text, It is highly recommended to present short note in a Table Title, so the table would be self-sufficient, i.e. it reviews the expression parameters on the 3h and 8h spans.

5)      I cannot immediately embrace what does the first column of S1 mean, others as well. It concerns other tables as well: column titles should be decoded within the table title/comment, especially in supplementary ones. See also 3) in this instance.

Author Response

Several notes on the manuscript,

  • it would be worth providing some putative (drug) targets (genes) at the conclusion section, if that’s not a closed info. The analysis of temporal dynamics of alternative splicing events would be also quite relevant within the scope of the paper.

Reply: We added specific information on ladostigil use in neonatal rodent microglia and the relevant references. Thanks for the suggestions. The issue of alternative splicing is of a great interest. However, it adds a new (and rather complex) dimension to the manuscript. As we were asked to simplify and shorten the manuscript, we did not include such splicing alteration analysis.

  • FDR (q-value) < 0.05 and >10 TMM - TPM  (s. 213).

Reply: Actually, we use TMM throughout as it provides normalized expression units that help to remove batch effects. Note that TMM is a between-sample method (in contrast to within-sample like TPM).  

Revised: We added a clarification in Materials and Methods.

  • I do not quite comprehend the ‘same’ category based on the Fig. S1. How could ‘same’ be in the first position? We’re talking of DEGs, so the genes cannot keep the initial expression rate. If it means that on the first 3 hours there’s not statistically significant expression change (?), then what does the ‘same-same’ term imply? The picture legend probably can be extended with some elaboration.

Reply: From the question of the referee, we understand that our notion of Up, Down and Same were not clear enough. We consider the FC1 (fold change for the early time of 3 hrs as cells with bzATP relative to untreated cells). The definition of FC2 is based on the results of FC1 (see revised Table 1). We included in the analysis all mapped RNA transcripts (note that thresholds on TMM and FDR values were included for highlighting the most significant expressed genes). The combinations yield 9 patterns of ‘combined trends’ (e.g., Up-Up, Up-Same) was better explained. 

Revised: We improve the definition of the trends (Revised Table 1). We added a new Figure 2A with all genes that compiled the 9-combined trends. We added the number of genes associated with each such pattern in revised Figure 2B. We added a clarification to the headers in Table S1 and Table S3.

  • Though the Table S1 acronyms are decoded in the text, It is highly recommended to present short note in a Table Title, so the table would be self-sufficient, i.e. it reviews the expression parameters on the 3h and 8h spans.
  • I cannot immediately embrace what does the first column of S1 mean, others as well. It concerns other tables as well: column titles should be decoded within the table title/comment, especially in supplementary ones. See also 3) in this instance.

Reply: Sorry for not being clear about the term “trend”. See above.

Revised: We added explanations as comments in Table S1 and Table S3. In revised Table 1 we explain the 9 combinations that are refer to combined trend. We added a comment in the text addressing the formal definitions.

Reviewer 2 Report

Zohar and colleagues have performed an exhaustive analysis of the transcriptional program induced in cultured mouse microglia by stimulation with bzATP and LPS. The authors indicate that the dual stimulation is necessary for a robust up-regulation of inflammatory genes. There is ample documentation of their findings in the figures provided in the body of the paper as well as the supplemental material. As a quantification of activation in microglia, this paper is a potential good addition to the literature.

1.     The title indicates that abundant non-coding RNAs define the inflammatory response, yet the ncRNA analysis not mentioned in the abstract, and the ncRNA influence is limited to the end of this long manuscript. In fact, most of the ncRNAs identified as affected by the stimuli are of unknown function, as stated in the manuscript. It is unclear why they chose to emphasize this in the title and then ignore it until the end of the manuscript.

2.     There are extensive figures, but not all are easily digestible. For example, Fig 6, which should be straightforward, is hard to read, as are others.

3.     The number of genes up and down regulated at different times is variable, making it difficult to follow the results section. The delineation of 3 and 8 hr timepoints is also confusing since the authors then concentrate (Fig 2) on the ‘up-up’ genes in this figure.

4.     Fig 3 PCA analysis should be explained further.

5.     In Fig 5, a good attempt at putting these changes in the context of infection/inflammation, perhaps only the ‘up-up’ genes should be included. Does it matter from the standpoint of pathophysiology if some genes are briefly upregulated?

6.     I was unclear how Fig 8 relates to the results (section 3.9) in terms of a comparison with published results.

7.     The supplemental figures are useful and extensive. If the authors step back and look at all of their figures, some within the supplemental files may be more informative than the ones within the manuscript.

In summary, this manuscript reports a useful catalogue of microglial genes affected by the stimuli. It can be refined to decrease the sheer volume of figures and results to emphasize the key points. The title does not reflect the bulk of the information provided and it should be changed to reflect what is in the paper. 

Author Response

Comments and Suggestions for Authors

Zohar and colleagues have performed an exhaustive analysis of the transcriptional program induced in cultured mouse microglia by stimulation with bzATP and LPS. The authors indicate that the dual stimulation is necessary for a robust up-regulation of inflammatory genes. There is ample documentation of their findings in the figures provided in the body of the paper as well as the supplemental material. As a quantification of activation in microglia, this paper is a potential good addition to the literature.

  1. The title indicates that abundant non-coding RNAs define the inflammatory response, yet the ncRNA analysis not mentioned in the abstract, and the ncRNA influence is limited to the end of this long manuscript. In fact, most of the ncRNAs identified as affected by the stimuli are of unknown function, as stated in the manuscript. It is unclear why they chose to emphasize this in the title and then ignore it until the end of the manuscript.

Reply: We agree that ncRNAs data is only a small addition to the major contribution on coding gene transcriptome in microglia culture.

Revised: We revised the title as suggested.

  1. There are extensive figures, but not all are easily digestible. For example, Fig 6, which should be straightforward, is hard to read, as are others.

Reply: This is an important comment. We tried to simplify some of the Figures and reduce their number.

Revised: Specifically, Revised Figures 2 and Figure 5 snd Figure 6 are simpler. We recreated Figure 6 and added the input gene list as Supplementary Table S5. We also removed the original Figure 6B. The revised Figure 6B is a simple version of the original Figure 8. In the revised version there are altogether 7 Figures. We eliminated the original Figure 8 that was based also on external report.

  1. The number of genes up and down regulated at different times is variable, making it difficult to follow the results section. The delineation of 3 and 8 hr timepoints is also confusing since the authors then concentrate (Fig 2) on the ‘up-up’ genes in this figure.

Reply: We realize that our notion of Up, Down and Same were not clear enough (it was also questioned by the other referee). We defined FC1 (fold change for the early time of 3 hrs as the results from cells with bzATP relative to untreated cells). The definition of FC2 is based on the results from FC1 (see revised Table 1). Thus, the notion Up-Up means genes that were substantially induced after 3 hr but also after 8 hr with respect to the baseline of expression in 3 hrs). The combinations of all these transcripts yield 9 ‘combined trends’ (e.g., Up-Up, Up-Same). We clarified the number of genes associated with each of the experimental settings (e.g., revised Figure 2).

Revised: We improve the definition of the trends (Table 1). We added Figure 2A with all genes that compiled the 9-combined trend (for ATP treatment). We added the exact number of genes associated with each of the combined trends. We also added a clarification to the headers in Table S1 and Table S3.

  1. Fig 3 PCA analysis should be explained further.

Revised: We simplified the PCA explanation (and removed the analysis that was represented in the original Fig. S2).

  1. In Fig 5, a good attempt at putting these changes in the context of infection/inflammation, perhaps only the ‘up-up’ genes should be included. Does it matter from the standpoint of pathophysiology if some genes are briefly upregulated?

Reply: Thanks, we agree that only Up-Up is relevant.  The rest of the analysis is less informative or interesting and was eliminated.

Revised: Revised Fig. 5. We developed the GO annotation enrichment to emphasize the strong signal of inflammation and cytokine release in all three branches of GO annotation.

  1. I was unclear how Fig 8 relates to the results (section 3.9) in terms of a comparison with published results n.

Reply: This Figure was removed as the information that was based on external published results.

Revised: In the revised Figure 6B we only include the section of the KEGG pathway based on our results.

  1. The supplemental figures are useful and extensive. If the authors step back and look at all of their figures, some within the supplemental files may be more informative than the ones within the manuscript.

Reply: Thank you for the suggestion. We revisited and changed accordingly.

Revised: (1) Only the section of the KEGG pathway based on our results is included in Revised Fig. 6B. (2) We included the supplementary in Figure 2A to visually explain the notion of the combined trend. (3) We removed the additional PCA-like analysis for the top 1000 genes to avowing confusion and to simplify the analysis. (4) We added revised Table S5 to list the input for the STRING analysis (that was redone). (5) We removed the original Figure 6B. (5) We added simple explanation to the headers of Table S1 and S3. (6) We eliminated the analysis of INF addition to microglia (original Figure 8B).

In summary, this manuscript reports a useful catalogue of microglial genes affected by the stimuli. It can be refined to decrease the sheer volume of figures and results to emphasize the key points. The title does not reflect the bulk of the information provided and it should be changed to reflect what is in the paper.

Reply: As suggested we change the title to fit the content of the manuscript.

Round 2

Reviewer 2 Report

The manuscript has been revised extensively. I do not have any additional comments.